# Bridging the gap between atomically thin semiconductors and metal leads

Xiangbin Cai [1,5], Zefei Wu [1,5], Xu Han [1,2], Yong Chen[1,3], Shuigang Xu[1], Jiangxiazi Lin[1], Tianyi Han[1], Pingge He [1], Xuemeng Feng[1], Liheng An[1], Run Shi [1,3], Jingwei Wang[1,3], Zhehan Ying[1], Yuan Cai[1], Mengyuan Hua[4], Junwei Liu [1], Ding Pan [1,2], Chun Cheng [3] & Ning Wang [1✉]

Electrically interfacing atomically thin transition metal dichalcogenide semiconductors (TMDSCs) with metal leads is challenging because of undesired interface barriers, which have drastically constrained the electrical performance of TMDSC devices for exploring their unconventional physical properties and realizing potential electronic applications. Here we demonstrate a strategy to achieve nearly barrier-free electrical contacts with few-layer TMDSCs by engineering interfacial bonding distortion. The carrier-injection efficiency of such electrical junction is substantially increased with robust ohmic behaviors from room to cryogenic temperatures. The performance enhancements of TMDSC field-effect transistors are well reflected by the low contact resistance (down to 90 $\Omega\mu m$ in $MoS_2$, towards the quantum limit), the high field-effect mobility (up to 358,000 $cm^2V^{-1}s^{-1}$ in $WSe_2$), and the prominent transport characteristics at cryogenic temperatures. This method also offers possibilities of the local manipulation of atomic structures and electronic properties for TMDSC device design.

[1] Department of Physics and Center for Quantum Materials, The Hong Kong University of Science and Technology, Clear Water Bay, Kowloon, Hong Kong, China. [2] Department of Chemistry, The Hong Kong University of Science and Technology, Clear Water Bay, Kowloon, Hong Kong, China. [3] Department of Materials Science and Engineering, Southern University of Science and Technology, Shenzhen 518055, China. [4] Department of Electrical and Electronic Engineering, Southern University of Science and Technology, Shenzhen 518055, China. [5] These authors contributed equally: Xiangbin Cai, Zefei Wu. ✉email: phwang@ust.hk

With a geometry similar to that of graphene, atomically thin transition metal dichalcogenide semiconductors (TMDSCs) possess many valuable properties that further broaden the two-dimensional (2D) materials playground[1–3]. However, in contrast to graphene, the electrical performance of atomically thin TMDSC field-effect transistors (FETs), such as the carrier mobility at cryogenic temperatures, is generally insufficient to explore many quantum transport properties or practical applications[4–6]. Electrically interfacing atomically thin TMDSCs with metal leads is inherently problematic because of undesired metal-semiconductor interface barriers. These include the tunnel and Schottky barriers as shown in Figs. 1a, b, respectively. Fermi-level pinning effects and Schottky barriers naturally arise when defect-induced gap states occur at the interface[7,8]. These interfacial barriers dramatically suppress the carrier-injection efficiency by capping the available carrier mobility and presenting large contact resistance in TMDSC FETs.

Great efforts have been made over the past decade to minimize the electrical contact barriers in TMDSC FETs, which can be grouped into two major categories: (I) direct metallization of the contact region through doping treatments[9–11] or microscale phase transformation[12–14], and (II) van der Waals (vdW) contacts by inserting tunnel-barrier layers into the

metal-semiconductor junction[15,16], or by using graphene/soft-landed metals as vdW-interfaced electrodes[17–23]. It is notable that through extra gating on the contact regions, graphene leads can eliminate Schottky barriers in few-layer MoS₂ FETs[17,18]. In this case, however, the inherent nature of vdW contacts still limits the charge carrier injection, because the large vdW gap (3–4 Å) between the metal lead and TMDSC surface acts as an additional tunnel barrier against the carrier flow.

An efficient carrier injection to the conduction or valence bands of TMDSCs requires effective orbital overlap or hybridization between the transition metal and the electrode atoms. For example, the conduction band minimum of monolayer TMDSCs arises mainly from the $d$-orbitals of transition-metal atoms[24], which are sandwiched by two layers of chalcogen atoms. Any carrier injection through the chalcogen surface of TMDSCs (i.e., across the vdW gap) is less efficient because the orbitals of chalcogen atoms contribute little to the energy band edges. Alternatively, an edge-contact strategy can offer the advantage of orbital hybridization to transition-metal atoms[25]. However, Fermi-level pinning effects caused by the dangling bonds of TMDSCs edges inevitably result in severe Schottky barriers, degrading the FET performance at low temperatures (Supplementary Fig. 1). Since rich quantum transport behaviors of fundamental importance, such as topological states and electron correlation effects, are completely perturbed by the thermal stimulation and the strong phonon scattering of atomically thin TMDSC channels at room temperature, probing fascinating condensed matter physics in TMDSCs requires the significant advance in their low-temperature contact performance.

To address these issues, we develop a local bonding distortion (LBD) strategy for realizing highly efficient electrical junctions with atomically thin TMDSCs down to the monolayer limit. Both atomic-resolution cross-section electron microscopy and Raman spectroscopy evidence the LBD as a nanoscale trigonal-prismatic-to-octahedral coordination change of metal-chalcogen polyhedra in the metal-TMDSC interface. The LBD region acts as a semi-metallic bridge between the metal lead and the pristine TMDSC channel, exhibiting robust ohmic behaviors from room to cryogenic temperatures. The LBD contacts significantly elevate the carrier-injection efficiency of TMDSC FETs, demonstrating the low contact resistance down to 90 Ωμm in 3L-MoS₂ and the high field-effect mobility up to 358,000 cm²V⁻¹s⁻¹ in 5L-WSe₂ with prominent transport characteristics at 0.3 K. This work not only paves the way to exploring the unconventional quantum transport properties in TMDSCs, but also provides a scheme in the local manipulation of atomic structures and electronic properties of TMDSCs. We note that there are pioneering contact improvement works by argon-plasma-induced T phase[14], but the energetic argon bombardment actually creates vacancies to stabilize the phase transition, which is different from our structurally seamless soft oxygen plasma method. And our LBD is not the conventional 1 T phase transition, but an intermediate structure towards tetragonal phases without vacancies or dangling bonds. Our work establishes an innovative approach to enhancing the orbital hybridization and carrier-injection efficiency in metal-TMDSC contacts in a highly controllable way, which have not been dealt with in previous reports.

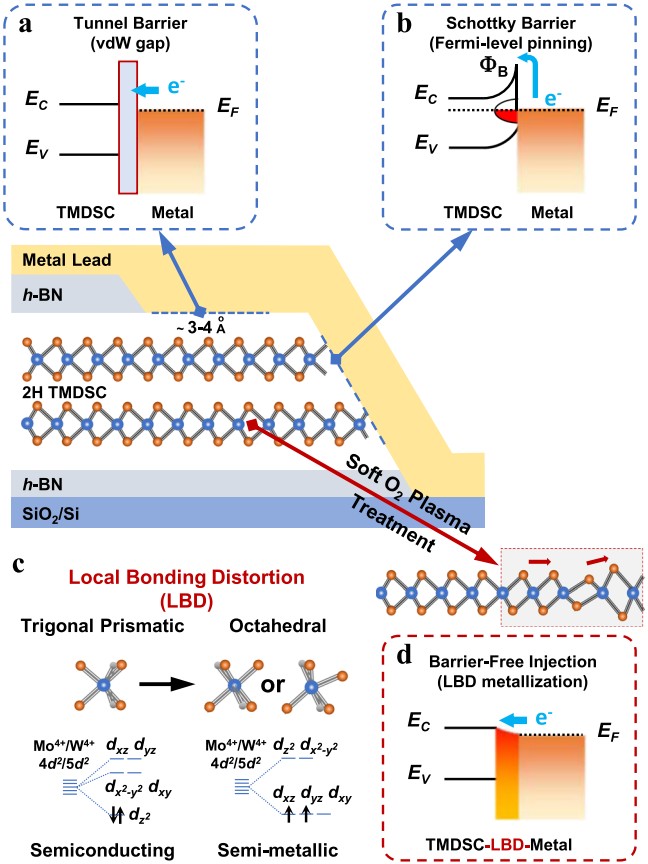

**Fig. 1 Different electrical interfaces between atomically thin TMDSCs and metal leads. a** Interfacial band alignment of the van der Waals (vdW) contact. **b** Interfacial band alignment of the Schottky-limited contact. **c** Atomic configurations of [TM-C₆] polyhedra and corresponding energy splitting diagrams of Mo/W d-orbitals before and after the local bonding distortion (LBD). **d** Interfacial band alignment of the contact utilizing LBD mechanism. $E_C$ and $E_V$ represent the energy of conduction and valence band edges in TMDSCs, respectively. $E_F$ indicates the Fermi-level energy in metal leads while $\Phi_B$ denotes the Schottky barrier height. Blue arrows suggest the routes for carrier injection, where e⁻ indicates the electron branch.

## Results

**Working mechanism of the LBD in different contact geometries.** First of all, we would like to introduce the basic mechanism of our LBD strategy. As shown in Fig. 1c, due to the trigonal-prismatic coordination of [TM-C₆] polyhedra in the H structure (in which TM stands for the transition metals, and C stands for

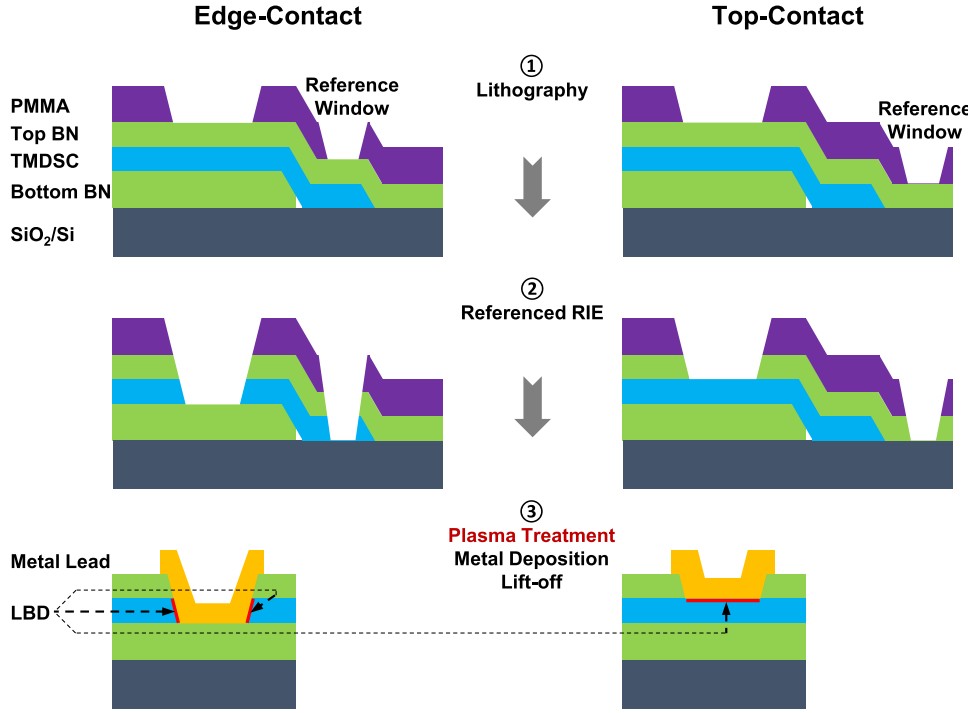

**Fig. 2 Mechanism of the referenced RIE technique.** Schematics showing the workflow of fabricating LBD contacts to BN-encapsulated TMDSCs in both edge- and top-contact geometries. After the soft oxygen plasma treatment, LBD appears on the exposed TMDSC edges or surface for the edge- and top-contact geometries, respectively.

the chalcogen atoms), the TM$^{4+}$ $d$-orbitals exhibit a close-shell electronic configuration (two $d$-electrons with opposite spins in one orbital) without itinerant electrons, resulting in semiconducting properties. In contrast, the octahedral coordination after distortion owns a rearranged energy splitting for the TM$^{4+}$ $d$-orbitals. The partial filling of two $d$-electrons with parallel spin into the three-fold degenerate orbitals provides both unoccupied states and itinerant electrons, resulting in the metallicity of the material[26]. The local distortion of [TM-C$_6$] polyhedra from trigonal-prismatic coordination to octahedral one modulates the electronic property into semi-metallicity, offering dispersed density of states (DOS) along the TMDSC energy band gap, and thus forms nearly barrier-free electrical interfaces as illustrated in the interfacial band diagram of Fig. 1d. Such lattice distortion is different from destructive defects and has been applied to engineer bulk materials' properties[27,28], while it has not been reported in TMDSCs yet. It is also different from the common H-to-T phase transition in TMDSCs by its nanoscale sizes and tunable work functions (Supplementary Fig. 11).

Technically, we chose pure oxygen (a chalcogen element) plasma to trigger the LBD in the contact interfaces of several representative 2D semiconductors, including MoS$_2$ and WSe$_2$. Schematics in Fig. 2 show the workflow of our referenced reactive ion etching (RIE) processes to fabricate LBD structures in both edge- and top-contact geometries. A thin poly(methyl methacrylate) (PMMA) mask is defined by e-beam lithography for the simultaneous RIE of the contact windows and the reference area. By monitoring the optical contrast of the reference area, either a top-BN/TMDSC/substrate region in the edge-contact geometry or a top-BN/substrate region in the top-contact geometry, the etching depth in contact windows is controlled precisely (the optical contrast of 300-nm-SiO$_2$/Si substrate will be totally different without 2D materials coverage). Followed by the tuned soft oxygen plasma treatment, LBD appears on the exposed TMDSC edges or surfaces for different contact geometries. We want to emphasize that the oxygen plasma treatment is configured

in a soft-landing manner to avoid generating defects in TMDSCs by paralleling the capacitively coupled electrostatic field along the sample surface, reducing the plate bias and lowering the input power. More details of the FET fabrication can be found in the Method section and step-by-step images in Supplementary Fig. 2. The LBD is reproducible once suitable conditions are set up. With reasonable proposals, we are pleased to provide devices utilizing LBD contacts for collaborative research.

**Atomic origin of the LBD contact.** Our LBD contact strategy works perfectly in the edge-contact geometry for exploiting its strong orbital overlap and hybridization advantages as demonstrated by 3L-MoS$_2$ FETs. In the top and cross-section views of Figs. 3a, b, after electrical measurements, a cross-section membrane from one of the main electrical contacts was lifted out for the electron microscopy observation. In the low-magnification annual dark field (ADF) image of the contact region (Fig. 3c), the elegant edge-cutting of the BN-MoS$_2$-BN stack by the referenced RIE process can be observed. The electrical interface between the 3L-MoS$_2$ edge and the deposited metal was observed at atomic resolution along the zig-zag direction, as shown in Fig. 3d. The atomic-resolution ADF image shows three main structural features of the plasma-induced LBD in the edge-contact region: (1) the LBD localizes within the ~1 nm wide edge of 3L-MoS$_2$; (2) both Mo and S atomic positions deviate from the ideal 2H phase on the left and form octahedral configurations; (3) terminated Mo atoms at the MoS$_2$ edge connect directly to the lead surface atoms with a closest distance of ~1.8 Å, which is much smaller than the normal vdW gap size (~3–4 Å) and similar to that of a typical chemical bond. This atomic-scale observation supports the strong orbital overlap and hybridization between transition-metal and electrode atoms in an edge-contact geometry.

The first layer of MoS$_2$ was further analyzed in Fig. 3e. Since the atomic structure on the left H region remains identical with the simulated one, there should be no damage or artifact

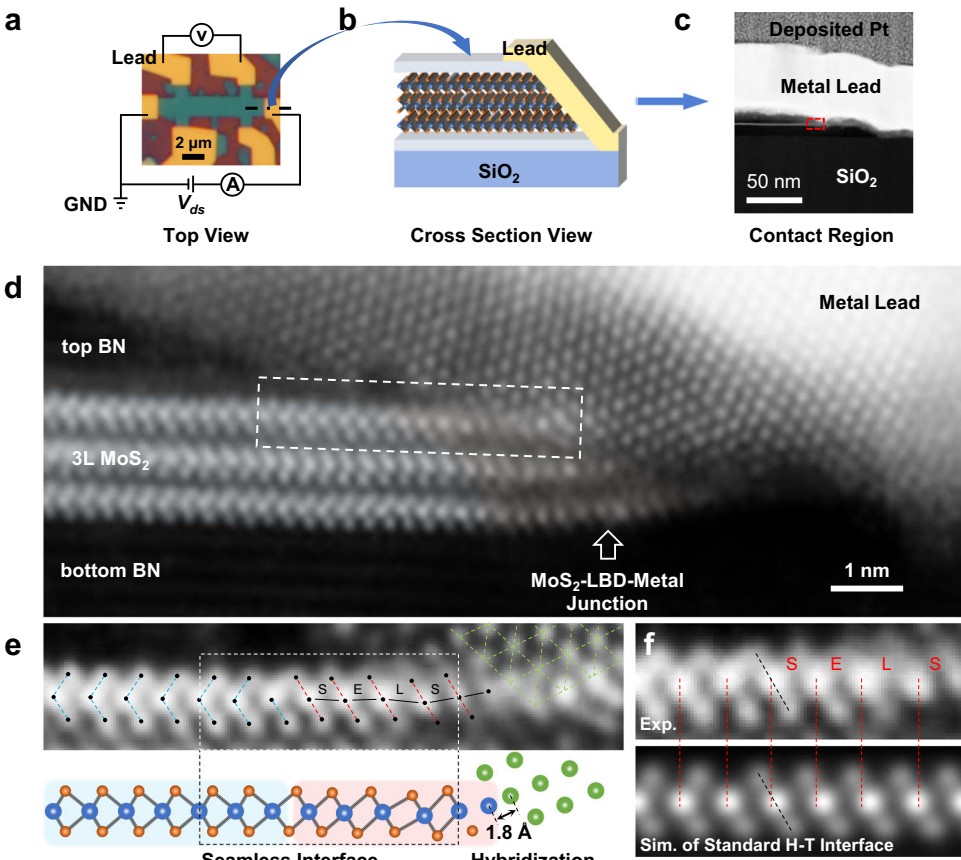

**Fig. 3 Atomic-scale observation of the LBD contact in MoS₂. a** Top-view optical image of a typical 3L-MoS₂ FET. The dash-dot line indicates the cutting position in the source, from which a cross-section membrane as illustrated in the schematic (**b**) was extracted using the focused ion beam (FIB) method. **c** The low-magnification ADF image of the contact region. The area in the red-dash rectangle is zoomed in as (**d**). **d** Atomic-resolution ADF image of the MoS₂-LBD-metal junction observed along the MoS₂ zig-zag direction. The first layer of MoS₂ is analyzed in (**e**) with the corresponding atomic model shown below. The overlaid blue, red and green dashed lines highlight the lattices of hexagonal MoS₂, octahedral MoS₂ and metal lead, respectively. **f** The Mo–Mo bond-length comparison (E equal, S short, L long) between the experimental measurement and the simulated standard structure. The red dashed lines mark the positions of Mo atomic columns in the experimental data while the black dashed lines indicate the S-Mo-S unit from which the distortion starts.

introduced by our microscopy observation. The safeness of our imaging conditions to exclude any e-beam artifacts is also demonstrated by the controlled imaging of pristine monolayer MoS₂ under the same conditions as used in the atomic imaging of device contact regions (Supplementary Fig. 13). Both H–T heterophase interfaces are structurally seamless without dangling bonds or defects, where only slight stretching or shrinking of some bond lengths is involved in the distortion. This 1-nm-wide structural change looks similar to the 1 T phase transition, but the Mo–Mo bond angle, length, and interlayer stacking order differ from those of the 1 T phase as compared with the simulated standard H–T interface in Fig. 3f. In specific, the Mo–Mo spacings in the experimental measurement stagger around the equal value in an ideal 1 T phase. The first unit cell in the termination shrinks a bit while the second one on the left elongates. The deviation of atomic positions from standard crystallographic sites as observed here should affect the electronic property through the electron-lattice coupling, which makes the DOS of such semi-metallic nanostructure dispersed away the theoretical work-function value of the ideal crystal. The energy dispersion of DOS thus allows the nearly barrier-free electron transport from the metal Fermi level to the TMDSC conduction band edge as illustrated in the interfacial band diagram of Fig. 1d.

By performing the electron energy-loss spectroscopy (EELS) analysis in this LBD contact region as shown in Supplementary

Fig. 3, we found oxygen existing in the distorted MoS₂ edge, which is consistent with the proposed formation mechanisms of LBD by the oxygen-substitution-induced bond rearrangements[29]. On one hand, the oxygen plasma transfers kinetic energy to treated samples to drive the bond distortion; On the other hand, the generated oxygen ions can substitute some chalcogen atoms during the soft O₂-plasma treatment and thus locally modifies the bond length, stabilizing the distorted structure. As depicted in Supplementary Fig. 4a, only stretching or shrinking of some bond lengths and angles is involved in the LBD, which in turn balances the size difference between chalcogen and oxygen atoms after the substitution. It is also reported that the incorporation of oxygen species may increase the carrier transmission in the electrical interface with metals[29,30]. The O₂-plasma-induced LBD was further confirmed by micro Raman spectroscopy (Supplementary Fig. 4b). The local distortion of [TM-C₆] polyhedra from the trigonal-prismatic coordination to octahedral ones modulates the electronic property into semi-metallicity, boosting the carrier-injection efficiency of electrical contacts and thus device performance as discussed below.

**Electrical performance of the LBD contact**. The 3L-MoS₂ FET interfaced by the LBD contacts shows excellent performance at both room and cryogenic temperatures. In Fig. 4a, b, linear

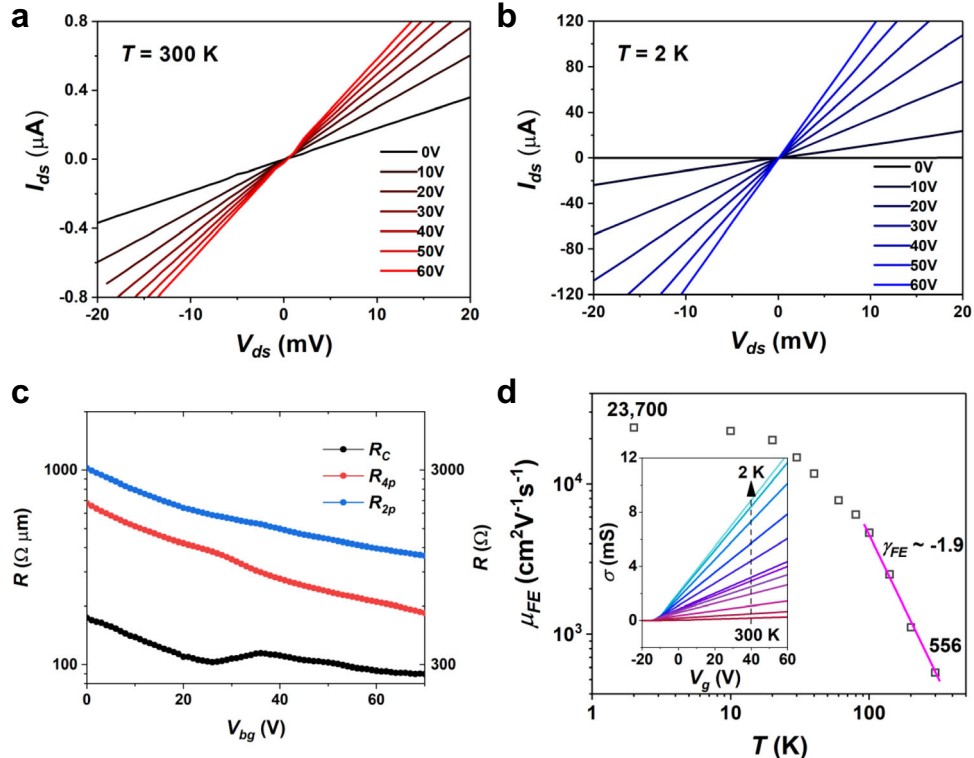

**Fig. 4 Electrical performance of the 3L-MoS₂ FET utilizing LBD contacts.** Two-probe $I_{ds}$–$V_{ds}$ curves at (**a**) room temperature and (**b**) 2 K, respectively. **c** Contact resistance, $R_c$, extracted from the resistance difference between two-probe and four-probe measurements as the function of gate voltages, where the four-probe resistance is scaled by $\frac{L_{out}}{L_{in}}$ as described in the Methods section and both resistance values in $\Omega \cdot \mu m$ and $\Omega$ units are plotted in the logarithmic scale. It can be seen that the contact resistance depends little on the back gating after the device is fully turned on. The off-state of this device locates below −10 V back-gate voltage due to the natural electron doping of MoS₂ as demonstrated in the inset of Fig. 4d. **d** Temperature-dependent field-effect mobility, $\mu_{FE}$, derived from the channel conductance as the inset shows, showing the strong phonon scattering effects of 3L-MoS₂ at high temperatures. $\gamma_{FE}$ represents the exponential temperature decay constant of the field-effect mobility.

two-probe output curves, $I_{ds}$–$V_{ds}$, can be observed at all gate voltages and well maintained down to 2 K, demonstrating the robust ohmic contact nature. The channel current at 2 K is about 200 times larger than that at 300 K, which suggests a little contribution from the contacts to the total device resistance. In Fig. 4c, the extracted contact resistance at 2 K decreases to around 270 $\Omega$ as the back-gate voltage increases, which is 90 $\Omega\mu m$ after considering the geometry factor of device channel. We notice that the quantum limit of the contact resistance is inversely related to the 2D charge carrier density ($n$), yielding $0.026/\sqrt{n} \approx 30\ \Omega\mu m$ when $n = 10^{13}\ cm^{-2}$, which suggests that our LBD contact quality is approaching the theoretical limit[4]. The channel conductance in the inset of Fig. 4d increases steadily by lowering the temperature, reflecting the gradually lifted phonon scattering in the MoS₂ channel. The extracted field-effect mobility shows high values of 556 $cm^2V^{-1}s^{-1}$ at room temperature and 23,700 $cm^2V^{-1}s^{-1}$ at 2 K with a phonon-scattering constant of around −1.9, confirming the strong phonon–electron interaction in atomically thin MoS₂.

This contact scheme combined with the top-contact geometry as discussed below also works well for monolayer MoS₂ as demonstrated in Supplementary Fig. 5. The metal-LBD-MoS₂ interfaces effectively lift the carrier-injection barriers in 1L-MoS₂ FETs, manifesting high-quality quantum-Hall detection, brilliant subthreshold swings, high field-effect, and Hall mobilities (9900 and 9200 $cm^2V^{-1}s^{-1}$, respectively) at cryogenic temperatures. Such high-performance 1L-MoS₂ devices made by the LBD method should facilitate future study on the electron–electron interaction effects at fractional quantum Hall states. The relatively suppressed carrier mobility for 1L-MoS₂ may be due to the

weaker screening effect of monolayer channels and thus increased scattering by intrinsic impurities, but the effectiveness of our LBD strategy for monolayer TMDSCs is clarified.

**Generality of the LBD contact.** Furthermore, we demonstrate the generality of the LBD strategy using 5L-WSe₂ in the more common top-contact geometry. The optical image of a typical top-contacted device is shown in Fig. 5a. The top-contact geometry is widely used as it provides a much larger carrier-injection area (as illustrated in Fig. 5b) than the edge-contact scheme. The precise removal of top BN to expose the pristine WSe₂ surface is realized by the referenced RIE process as illustrated in Fig. 2. From the low-magnification ADF image of the contact region in Fig. 5c, a uniform top-contact area can be identified, except for some top BN islands that remained due to local thickness differences. After the soft oxygen plasma treatment, the topmost WSe₂ layer within the contact window went through the LBD without affecting the layers beneath as shown by the atomic-resolution ADF images of metal-WSe₂ electrical junction in Fig. 5d. This LBD metallization exhibits complicated changes of the W–Se bonding without any crystallographic defects (e.g. vacancies or dangling bonds). According to the 2D-Gaussian-fitted W and Se atomic column positions overlaid in the image (Supplementary Fig. 6 for the image-processing details), the newly formed structure turns out to be a mixture of 1 T and 1 T' octahedral derivatives. Since the nanoscale T/T'-mixed structure does not create defects and thus can be considered as a distorted crystalline structure. Such distorted WSe₂ layer acts as a semi-metallic bridge between the

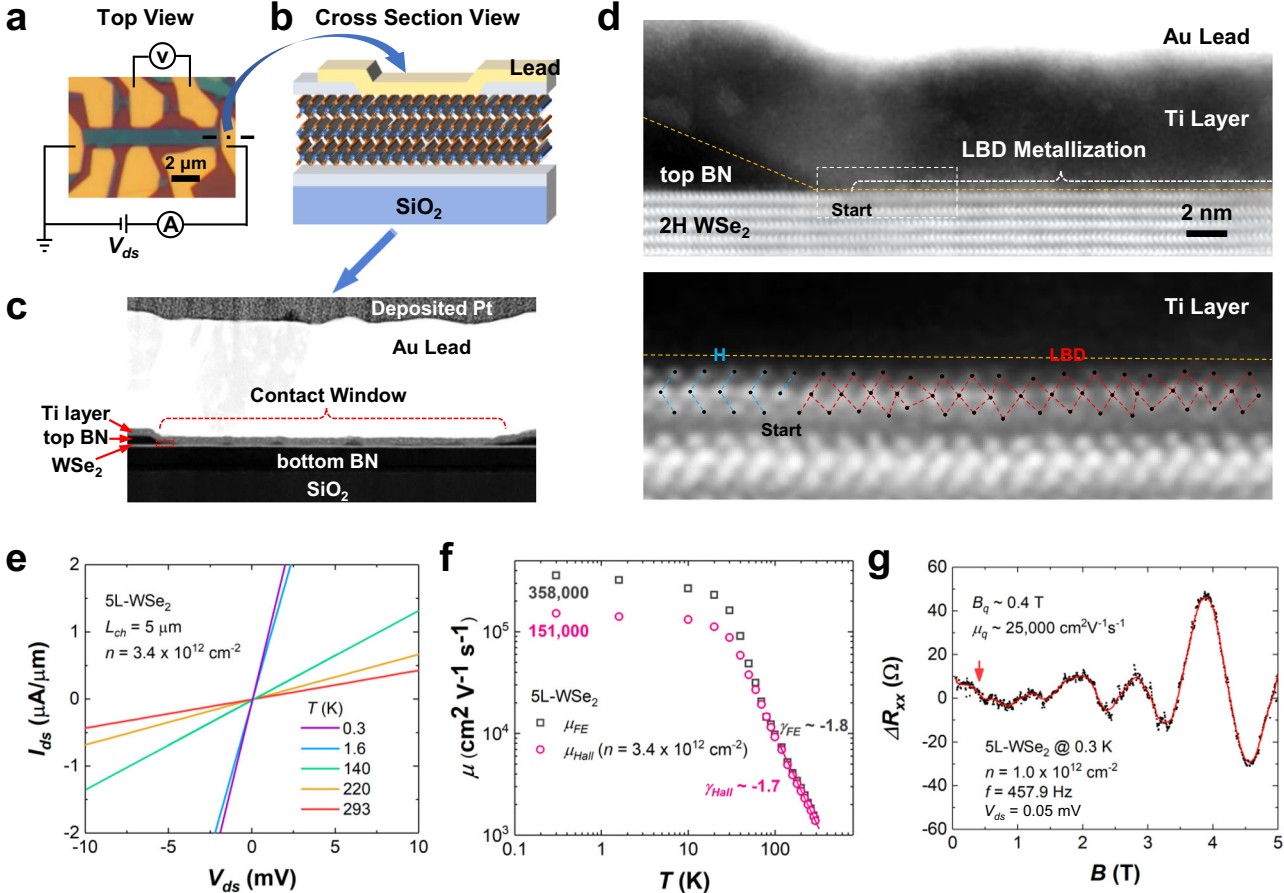

**Fig. 5 Generality of the LBD strategy. a** Optical image of a typical 5L-WSe$_2$ FET utilizing LBD contacts. The dash-dot line indicates the cutting position in the source, from which a cross-section membrane was lifted out as illustrated in **b**. **c** The low-magnification ADF image of the contact region. The area in the red-dash rectangle is zoomed in as **d**. **d** Atomic-resolution ADF image of the WSe$_2$-LBD-metal junction observed along the WSe$_2$ zig-zag direction. The orange dashed lines show the bottom of Ti layer. The LBD-starting region (white dashed rectangle) is analyzed below with the contrast re-surveyed, and the atomic model overlaid to clarify the distorted bonding. The overlaid blue and red dashed lines highlight the lattices of hexagonal WSe$_2$ (H) and octahedral WSe$_2$ (LBD), respectively. **e** Tow-probe I$_{ds}$-V$_{ds}$ curves at varied temperatures. **f** Log-scale plot of field-effect and Hall mobilities as the function of temperatures. $\gamma_{FE}$ and $\gamma_{Hall}$ represent the exponential temperature decay constants of the field-effect and Hall mobilities, respectively. **g** Shubnikov de Haas (SdH) oscillations of the longitudinal channel resistance at low magnetic field strength down to ~0.4 T, indicating the high quantum mobility. The solid red line is averaged from the nearest 25 data points and the red arrow indicates the prominent onset position of SdH oscillations at ~0.4 T.

metal lead and the pristine WSe$_2$ channel, leading to a nearly barrier-free carrier injection, similar to the cases in MoS$_2$.

Linear output characteristics are observed across a wide temperature range as shown in Fig. 5e, proving the robust ohmic contact nature. Although the contact resistance of 700 Ωμm is larger than that in the edge-contacted MoS$_2$ due to the inevitable vdW gap in the top-contact geometry (see Supplementary Fig. 7 for extended data from this top-contacted WSe$_2$ FET), high field-effect ($\mu_{FE} \sim$ 358,000 cm$^2$V$^{-1}$s$^{-1}$), Hall ($\mu_{Hall} \sim$ 151,000 cm$^2$V$^{-1}$s$^{-1}$) and quantum ($\mu_q \sim$ 25,000 cm$^2$V$^{-1}$s$^{-1}$) carrier mobilities are achieved at 0.3 K, as the temperature plots in Fig. 5f and the prominent Shubnikov de Haas (SdH) oscillations in Fig. 5g demonstrate. Although the quantum mobility is a rough estimation from the oscillation onset at low magnetic fields, the derived value should be the lower bound, because the electrical turbulence from both the measurement system and the external environment will bury the real onset of the oscillation amplitude smaller than the noise level. Theoretically, there is $\mu_{FE} > \mu_{Hall} > \mu_q$ for the same sample and temperature. The field-effect mobility is always higher than the Hall mobility of the same conditions because of $\mu_{FE} = \mu_{Hall} + n \frac{d\mu_{Hall}}{dn}$, where higher carrier density can screen the impurity scattering and results in $\frac{d\mu_{Hall}}{dn} > 0$. On the other hand, since the quantum scattering

time is always shorter than the transport scattering time, quantum mobility should fall below the corresponding Hall mobility. The quantum mobility is an evaluation of the system's quality for observing quantum phenomena (see more detailed quantum transport data of such high-mobility WSe$_2$ device in Supplementary Fig. 14). No hysteresis effects were observed when scanning gate voltage forwards and backwards as shown in Supplementary Fig. 7c, which confirm there are no chargeable states caused by the LBD, in consistence with the atomic microscopy observation of no dangling bonds or vacancies formed in the metal-TMDSC junction. Furthermore, the reproducible contact quality after the long-time storage as shown in Supplementary Fig. 7d and the competitive performance of multiple devices with the same contact geometry and different channel thicknesses shown in Supplementary Fig. 8 demonstrate the excellent reproducibility and practicality of our LBD contact strategy for TMDSCs.

## Discussion

The high performance of the electrical interfaces formed by the LBD method is attributed to the octahedral distortion of TMDSCs. Unlike a perfect H–T interface, in which the work functions of bulk 1 T and H phases are mismatched, various sizes

**Table 1 RIE conditions used in the contact fabrication.**

| Process | RF power and frequency | Gas composition | Pressure and flow rate | Substrate bias | Duration |
|---|---|---|---|---|---|
| Etching | 200 W 13.56 MHz | $CHF_3:O_2 = 10:1$ | Base pressure 6 mTorr; Flow rate 40:4 sccm | 20 W | [a]5-15 s |
| Plasma treatment | 100 W 13.56 MHz | $O_2$ 100% | Base pressure 6 mTorr; Flow rate 40 sccm | 0 W | 10 s |

[a]The etching duration depends on the thickness of the top BN layer in the top-contact geometry or BN/TMDSC/BN stack in the edge-contact geometry.

of distorted T/T' derivatives induced by the oxygen plasma are semi-metallic, offering dispersed energy states around its work function. These energy states may play an important role in the orbital hybridization between transition metals and lead atoms, thus enhancing the carrier-injection efficiency. This mechanism is supported by the fact that the polarity of the LBD interfaced FETs is contact-metal dependent. For example, Ti/Au leads can access the conduction band edge (n-type) while Pd leads can access the valence band edge (p-type) of $WSe_2$ (Supplementary Fig. 9). The DOS of the LBD region is distributed along the energy band gap of TMDSCs, easily coupled with different work functions of metal leads. We also measured the resistance-temperature relationship of $1L-MoS_2$ flake after treated by the same soft oxygen plasma process as that used to fabricate LBD contacts (Supplementary Fig. 10), where the LBD-induced semi-metallicity and the strong electron-phonon coupling are proved. For the standard bulk T–H interface, however, the device polarity is independent of contact metals used[12] since the work function of 1 T phase is fixed, resulting in limited accessibility to the energy bands of TMDSCs.

We also conducted transmission line experiments[31] on n-type $MoS_2$ using LBD edge contacts, n-type and p-type $WSe_2$ using LBD top contacts as shown in Supplementary Fig. 12. It can be seen that the obtained best contact resistance is $85 \pm 15\ \Omega\mu m$ in edge-contacted $3L-MoS_2$, which is very close to our estimation of $90\ \Omega\mu m$. For n-type top-contacted $WSe_2$, the contact resistance falls within $100–200\ \Omega\mu m$ across ranges of carrier densities and temperatures. As a complementary measurement, in Supplementary Fig. 15, we demonstrate the contact resistance extraction of LBD-contacted $MoS_2$ and $WSe_2$ devices by comparing the two- and four-terminal resistance, showing consistent contact resistance values at room temperature. In principle, the contact resistance is dependent on the carrier density and the temperature. The higher carrier density and lower temperature lead to the better contact resistance, which may be explained by the fact that the impurity scattering in channels or contact interfaces can be screened by clouded carriers and suppressed by lowered temperature. This correlation in top-contacted p-type $WSe_2$ is more prominent.

In conclusion, we demonstrate a proof-of-concept LBD contact strategy by the soft oxygen plasma treatment, which is flexible for both edge- and top-contact fabrication schemes, to construct high-quality electrical junctions between metal leads and different TMDSCs, down to their monolayer limit. The excellent performance of such electrical contacts is structurally understood by the cross-section electron microscopy observation of metal-semiconductor junctions from practical devices. By exploiting such high-quality contacts, we have achieved robust ohmic behaviors, the low contact resistance (down to $90\ \Omega\mu m$ in $3L-MoS_2$), the high mobilities (up to $358,000\ cm^2V^{-1}s^{-1}$ in $5L-WSe_2$) and prominent transport characteristics. Since the electrical contact quality at cryogenic temperatures has been considered as the main factor constraining the study of unconventional quantum transport properties in TMDSCs, our LBD contact strategy should facilitate high-quality TMDSC device fabrication for quantum physics study. Importantly, this method is fully compatible with the clean-room processes for scalable electrode integration and may shed light on the future vdW device design.

## Methods

**Materials**. TMDSC crystals were bought from the website www.2dsemiconductors.com while the hexagonal BN (grade A1) was bought from the website www.hqgraphene.com.

**FET fabrication**. The sandwich structure of top BN (5–8 nm), few-layer 2H-$MoS_2$/$WSe_2$, and bottom BN (12–20 nm) was assembled by the dry pickup-transfer technique in a glove box. Following the electron-beam lithography patterning of a thin poly(methyl methacrylate) (PMMA, A5) mask using Raith eLiNE, the heterostructure was first shaped into Hall-bar structure by the reactive ion etching (RIE) in STS Pro using the gas mixture of $CHF_3$ and $O_2$ (40:4 sccm). Repatterning a PMMA mask then defined the contact windows and a reference area for the referenced RIE as illustrated in Fig. 2. Before the deposition of metal leads, the TMDSCs were exposed to the pure oxygen-plasma flow for around 10 s in a soft-landing manner, which was realized by paralleling the electrostatic field along the sample surface, reducing the plate bias and lowering the input power. The non-destructive metal deposition[22], i.e. critically high vacuum of $3 \times 10^{-7}$ Torr and slow deposition rate of 0.2 Å/s, was performed by Peva 450E right after the oxygen plasma treatment to minimize the influence from the metal deposition process and seal the contact windows.

**RIE techniques**. The etching and plasma treatment were performed using the RIE system in a combination of parallel plate and inductively coupled plasma (ICP), model STS Pro, under the conditions as shown in Table 1. The ICP can generate a high-density source of ions while the separate RF bias in parallel plate can create directional electric fields near the substrate. In our soft oxygen plasma treatment, the vertical substrate bias is tuned to ~0 to reduce the kinetic energy of oxygen ions for complete reaction with samples and also to avoid possible etching effects.

The plasma treatment is reproducible and practical once suitable conditions are set up. As shown in Supplementary Fig. 8a, the contact resistance and field-effect mobility of multiple back-gated $3L-MoS_2$ and $5L-WSe_2$ FETs using the LBD contacts show similar high quality. The device deviation is reasonable considering the fabrication error by human handling and laboratory facilities. We notice that $WSe_2$ FETs show a larger device-to-device variation, which may be attributed to the higher mixed state of octahedral derivatives in LBD-treated $WSe_2$ than the situation of $MoS_2$.

**Electrical measurements**. The DC $I_{ds}-V_{ds}$ curves were measured in the two-probe configuration using Keithley 6430 while the channel conductance ($\sigma$) and magneto-transport measurements were conducted in the four-probe configuration using lock-in techniques (Stanford Research 830 as the amplifier and DS 360 as the function generator). The cryogenic station from Oxford Instruments was used to provide cryogenic temperatures down to 1.4 K and magnetic fields up to 15 T. A home-modified He3 holder provides a stable 0.3 K environment for ~24 h per regeneration.

Different types of carrier mobilities were calculated to assess the contact performance, including field-effect, Hall and quantum mobilities. The field-effect mobility ($\mu_{FE}$) values were obtained according to $\mu_{FE} = \frac{1}{C}\frac{\Delta\sigma}{\Delta V_g}$, where $C$ is the gate capacitance (from serially-connected capacitors of the 300 nm $SiO_2$ and the 12–20 nm bottom BN layer for back-gated devices), which was accurately determined by the Hall effect measurements of carrier density versus gate voltage relationship. The $\frac{\Delta\sigma}{\Delta V_g}$ is the slope of channel conductance ($\sigma = \frac{GL}{W}$, where $G$ is the four-probe conductance, $L$ is the channel length and $W$ is the channel width of inner probes) versus gate voltage at the linear region. The Hall mobility ($\mu_{Hall}$) values were obtained according to $\mu_{Hall} = \frac{\sigma}{ne}$, where $n$ is the carrier density derived from the Hall coefficient, $e$ is the elementary charge. The quantum mobility ($\mu_q$) value was derived from the lowest perpendicular magnetic field strength $B_q$ where SdH oscillations onset, as $\mu_q = \frac{1}{B_q}$. The exponential temperature decay of both field-effect and Hall mobilities follows $T^\gamma$, where $\gamma$ indicates the phonon scattering strength.

The contact resistance was extracted from the resistance difference between two-probe and four-probe measurements according to $2 * R_c = (R_{2p} - \frac{L_{out}}{L_{in}}R_{4p}) * W$, where $R_{2p}$ and $R_{4p}$ are the two-probe and four-probe resistance, respectively; $L_{out}$ and $L_{in}$ are the distance of outer probe pair and inner probe pair in the four-probe configuration, respectively.

**Cross-section sample preparation**. Focused ion beam (FIB) techniques using FEI Helios G4 UX were applied to prepare the cross-section membranes from FET contact sites, after electrical measurements. Before ion milling processes, a low-energy (2 kV) electron beam was used to identify the exact contact positions and deposit 500-nm-thick Pt protecting straps on the top of metal leads. Then thicker Pt protecting straps (1.5 μm thick) were deposited using the ion beam. After lifting out a 1-μm-thick plate from the chosen contact site to a copper finger, step-by-step thinning processes using 30, 5, 2, 1, and 0.5 kV ion beams well minimized the beam damage to the 2D-material heterostructures. Finally, an ex-situ 500 V argon-ion beam shower of 30 s was used to further clean both membrane surfaces. According to the low-loss electron energy-loss spectroscopy (EELS), the typical sample thickness at the contact interface is $12.3 \pm 2.5$ nm, which guarantees atomic-resolution imaging and elemental mapping at 60 kV acceleration voltage.

**STEM characterization**. JEOL JEM ARM 200 F (ACCELARM) equipped with a cold field-emission gun and the ASCOR fifth-order probe corrector was utilized for the aberration-corrected scanning transmission electron microscopy (ACSTEM) study of the contact interface at atomic resolution under 60 kV acceleration voltage. Imaging and spectroscopy were performed using 32 mrad convergence semi-angle and 24–60 pA probe current for the optimal information transfer and minimal electron irradiation. Our imaging conditions of low acceleration voltage of 60 kV, low electron irradiation dose of $\sim$7.5–20.0 Me nm$^{-2}$, and room temperature remain far away from the threshold to cause electron-induced structure change[32]. The collection semi angles for annular dark field (ADF) imaging was set to 68–200 mrad and the Gatan Enfinium spectrometer was used for the acquisition of energy-loss spectra under the dual EELS mode.

**Image analysis**. The atomic positions in model drawings of Fig. 3e and Fig. 5d were extracted by the 2D-Gaussian fitting of normalized intensities of atomic columns in the HAADF images. We show the data-processing workflow and distortion-analysis examples in Supplementary Fig. 6. The method can provide picometer precision for the fitted atomic positions when using an aberration-corrected STEM. Based on the extracted atomic positions, the LBD of oxygen-plasma-treated $MoS_2$ and $WSe_2$ can be identified by bond-length/angle variations unit cell by unit cell in comparison to the simulation of standard structures.

**Image simulation**. Multislice codes by Earl J. Kirkland were used for the simulation of atomic-resolution ADF images to handle the dynamical scattering processes between matter and transmitted electrons[33]. 2H, 1 T, and 1 T' structure files were downloaded from the American Mineralogist Crystal Structure Database. An electron probe of experimental conditions was focused on the crystal surface and the scattered electrons within 68–200 mrad collection semi angles were integrated. Twenty frozen phonon configurations were applied to represent the thermal diffuse scattering effects at room temperature. The effective source size effect was included by convolving simulated images with a 2D-Gaussian function of 0.8 Å FWHM.

**DFT calculations**. The plane-wave pseudopotential method implemented in the Quantum ESPRESSO package (version 6.1)[34] was used to calculate electronic structures. The SG15 Optimized Norm-Conserving Vanderbilt (ONCV) pseudopotentials were used here[35,36]. Fully relativistic pseudopotentials were employed in the calculations with spin-orbital coupling[37]. The kinetic energy cutoff for plane waves was 60 Ry. The convergence thresholds for energy, force, and stress were $10^{-5}$ Ry, $10^{-4}$ Ry/Bohr, and 50 MPa, respectively. The multilayer structures were obtained from full relaxation with the van der Waals functional optB88[38]. PBE exchange-correlation functional was adopted to calculate electronic structures[39].

## Data availability

Relevant data supporting the key findings of this study are available within the article and the Supplementary Information files. All raw data generated during the current study are available from the corresponding author upon reasonable request. Source data are provided with this paper.

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

## Acknowledgements

The work is partially supported by the National Key R&D Program of China (2020YFA 0309600), the Hong Kong Research Grants Council (Projects No. AoE/P-701/20, C7036-17W, 16305919, 16303720 and C6025-19G), and the William Mong Institute of Nano Science and Technology. C.C. thanks the funding support from the National Natural Science Foundation of China (Grants No. 91963129 and 51776094).

## Author contributions

N.W. conceived and directed the project. Z.W., X.C., Y.Ch., S.X., J.Lin., T.H., X.F., and L.A. conducted the device fabrication and electrical measurements. X.C. performed the FIB sample preparation and ACSTEM characterization. X.H., J.Liu., and D.P. carried out the DFT calculations. R.S., P.H., and J.W. grew the $MoS_2$ monolayer crystals. Z.Y. and Y.Ca. provided the technical support of TEM facilities. M.H. and C.C. assisted with electrical measurements. X.C. and N.W. wrote the paper. All authors participated in the data analysis, result discussion, and manuscript preparation.

## Competing interests

The authors declare no competing interests.
