## [Peer Review File · Nature Communications]

Bridging the gap between atomically thin semiconductors and metal leadsREVIEWER COMMENTS

Reviewer #1 (Remarks to the Author):

The authors claim that they have obtained nearly barrier-free metal-TMD contact with high carrier-injection efficiency and low contact resistance, R_c . The contact is realised by introducing octahedral distortion at the interface and hence enhancing the orbital hybridization. The distortion is induced by soft oxygen-plasma treatment. The distortion is supported by atomic-resolved ADF imaging and corroborated by Raman investigation. The authors investigated both edge and top contacts, demonstrating promising electrical performance at both RT and cryogenic temperatures. The manuscript is well written and the results are quite convincing.

Metal contacts are one of the key limiter to TMD devices (e.g. large contact resistance, residue and structural damage from lithographic process, adsorption, current crowding, etc.) Several methods have been proposed to remove this obstacle. Chemical induced phase transition has been demonstrated to lower R_c (e.g. Nature Mater 13, 1128–1134 2014), but this report realised a much lower R_c – perhaps the authors can comment on this. Encapsulating BN has been used previously to reduce extrinsic scattering (e.g., Nature Nanotech 10, 534–540, 2015), but the technique reported in this work enables efficient contact. The O₂ plasma was also extensively exploited in TMD device fabrication (e.g. etching in 2D Mater. 3, 021007 2016) and functionalisation (Science advances 4, eaao5031, 2018), while this work establishes an innovative approach to realising bond distortion in a controlled way to enhance the contact. Some other methods aim to reduce the damage introduced in metal deposition (e.g. Nature 557, 696–700 2018; Nano Letters, 1416, 2018), while none of them deals with the orbital hybridization – perhaps the authors can comment on the metal deposition method they used. Considering these merits, this referee considers this work represents a substantial technological advance over those previous studies for enhancing TMD transistors.

However, the referee has some minor concerns:

- As the key techniques, the RIE etching and plasma treatment are not described with sufficient details. The methodologies given in the manuscript cannot be followed and reproduced by other labs. The effectiveness of these methods are not described sufficiently in the manuscript. To demonstrate the reproducibility and practicality of the method, it is crucial to establish the controllability and precision of these methods.
- Raman results need further elaboration and clarification – spot size etc. Please explain how the 1nm distortion would cause such significant changes.
- Are there any observable distortion in the second layer of the WSe₂ sample? Please comment on this and explain its significance.
- The role of oxygen in the treated samples need to be clarified.

Reviewer #2 (Remarks to the Author):

Report for “Bridging the gap between atomically thin semiconductors and metal leads” by X. Cai et al.

The authors report on local bonding distortion (LBD) for good contacts between TMDSCs and metals. The impressive performances of the authors’ devices, such as the ultralow contact resistance (down to 90 ohm μm) and ultrahigh FET mobility (up to $358,000 \text{ cm}^2\text{V}^{-1}\text{s}^{-1}$), could be reported somewhere for 2D community, but this reviewer does not recommend this

manuscript for publication in Nature Communications by the following reasons.

1. Novelty issue:

As the authors wrote, microscale phase transformation (ref. 12, 13) has been reported for contact engineering. Plasma-treated phase engineering for contact has been also reported in other journals (not cited in this manuscript, for example JACS 139, 10216 (2017)).

The authors claimed, "It is also different from the common H-to-T phase transition in TMDSCs by its nanoscale localization." in page 5, but all the contact engineering would include randomly ordered nanoscale localization (or atomically defined contact between TMDSCs and metals) and edge structures.

Thus, this work could be considered an extension of previous works, and this reviewer judges that the manuscript lacks the critical novelty for nature communications.

2. Concerning the nanoscale localization mentioned in page 5, this reviewer believes that Raman spectroscopy shown in Fig. S4 should not be observed in the 1 nm-wide edge in Fig. 3. The authors better add more explanations.

3. The contact resistance of '90 ohm μm ' needs more supporting data. At zero gate bias, the current should be at the 'off-state' (e.g., we see such 'off state' in Fig. S7C). However, the y-axis intersection is 500 ohm, which is only five times the value at $V_g=60$ V. The lack of 'off-state' at zero gate bias may indicate some technical issues in the measurement.

4. The authors claimed that the slight deviation of atomic position makes an ideal contact like Fig. 1D. This is contrasted with our intuition. Such disordered atomic arrangements normally make Fermi level pinning to fix a Schottky barrier. Accepting the authors' explanation, how could the atomic deviation be well-adjusted for such an ideal contact? This might be an extraordinary coincidence. One possible explanation would be that a (vdW) tunneling barrier is formed by oxygen compounds for the improved contact.

5. The graphene edge contact has been found to be about '100 ohm μm ', and the edge contact in this work shows a better value than the graphene result. Given that the edge contact coupling length in graphene is shorter (~ 1 Å) than that of MoS₂ in this work (1.8 Å), the upper bound (90 ohm μm) in this work is too low. The authors show a Hall bar geometry in Fig. 5A. Have the authors conducted 4-probe measurements to precisely estimate the contact resistance?

Reviewer #3 (Remarks to the Author):

Cai et al. reported on the fabrication and characterisation of Hall bar-like TMDs (2H-MoS₂ and 2H-WS₂) field-effect devices with low apparent contact resistance. The authors claimed that the oxygen plasma treatment of the contact areas immediately before metalisation results in the contact resistance lower than the best values reported so far (see *Nature* **593**, 211 (2021)). The reduction of the contact resistance is attributed to the formation of the highly electrically conducting 1T phase of TMD, presumably under mild plasma treatment. The presence of lattice distortions in the contact area, which resembles the 1T phase, was observed by the TEM data for both materials used in the study. Similarly to the previous reports (*Nature Mater.* **13**, 1128 (2014)), the authors attributed the low apparent contact resistance, found from the two-terminal measurements at the highest possible charge carrier concentration, to the 1T-like lattice distortions expected to have metal-like electrical conductivity. Although the idea of crystal phase engineering to achieve low contact resistance is not new, the fabrication approach involving plasma treatment reported by Cai et al. is more practical and has significant technological potential.

However, in my opinion, the claims made in this work are very premature, and much additional work has to be done to prove the relation between the plasma-induced lattice distortions and

contact resistance as well as its actual value. Below are my comments and recommendations concerning this work:

1) The proper contact resistance measurement is absent from this work. The transmission line experiment should be carried out for each contact geometry (edge-contact and top-contact) as a function of channel carrier concentration and temperature. It is insufficient to quote the two-terminal channel resistance at 2 Kelvin to estimate the contact resistance in the entire range of possible parameters. A separate assessment of the situation is also needed for the *p*-type channel material, such as MoSe₂ or WSe₂.

2) Regretfully, the authors did not disclose the details of the 1T phase preparation procedure (soft oxide plasma treatment), which seems to be the main novelty of this work. For this work to be publishable, the authors must ensure that the description of the plasma treatment procedure is discussed in the smallest detail to be reproduced by the others. In particular, the following information should be provided:

- a) model and type of the RIE plasma system,
- b) RF power and frequency,
- c) gas mixture composition,
- d) pressure and flow rate,
- e) observed substrate bias,
- f) treatment duration.

Any other details/tips/hints should also be presented.

3) Some additional work with TEM should be done to establish the mechanism of plasma-induced structural transition and eliminate the possible experimental artefacts caused by the e-beam irradiation. For example, Lin et al. (*Nature Nanotech.* **9**, 931 (2014)) reported the e-beam induced 2H-to-1T transition in MoS₂. So, one has to ensure that the observed lattice distortion results from the plasma treatment and is not triggered by the e-beam exposure during TEM imaging.

4) Apart from the essential contact resistance characterisation, a more detailed transport study of the Hall bar devices needs to be done. On the one hand, it seems that the WS₂ device with nominally higher carrier mobility possesses minimal signs of quantum oscillations. On the other, a more modest MoS₂ device exhibits well-pronounced QHE steps. This inconsistency has to be explained.

Before this essential work is done, reporting the apparent low-contact resistance cannot be considered reliable. However, if the initial clues presented in the manuscript are proven correct, the impact of this work on the 2D material research and possible future application of TMDs in microelectronics will be significant.

Considering the above comments, I cannot recommend this work in its present form for publishing in either Nature Communication journal or any other Nature partner journal. I would strongly encourage the authors to continue working on this problem to prove the validity of the fabrication method by confirming the reproducible low-contact resistance to different materials and contact geometries and achieve a better understanding of its origin.

(Note: responses to reviewers' comments are in blue, changes made to the manuscript or the supplementary materials are highlighted by yellow background in respective documents and copied here in red for reference.)

Reviewer #1 (Remarks to the Author):

The authors claim that they have obtained nearly barrier-free metal-TMD contact with high carrier-injection efficiency and low contact resistance, R_c . The contact is realized by introducing octahedral distortion at the interface and hence enhancing the orbital hybridization. The distortion is induced by soft oxygen-plasma treatment. The distortion is supported by atomic-resolved ADF imaging and corroborated by Raman investigation. The authors investigated both edge and top contacts, demonstrating promising electrical performance at both RT and cryogenic temperatures. The manuscript is well written, and the results are quite convincing.

Metal contacts are one of the key limiter to TMD devices (e.g. large contact resistance, residue and structural damage from lithographic process, adsorption, current crowding, etc.). Several methods have been proposed to remove this obstacle. Chemical induced phase transition has been demonstrated to lower R_c (e.g. Nature Mater. 13, 1128-1134, 2014), but this report realized a much lower R_c - perhaps the authors can comment on this.

Response: Thank you so much for the positive evaluation on this work. In the mentioned work (Nature Mater. 13, 1128-1134, 2014), the chemical induced phase transition occurred across micrometers of few-layer MoS₂, so the electrode was formed on bulk 1T MoS₂. The work function of bulk 1T MoS₂ mismatches the electron affinity of 2H MoS₂, leading to the Schottky-barrier-limited contact and larger contact resistance (Phys. Rev. B 96, 165402, 2017). When the size of 1T phase decreases (~1 nm in our cases), its work function lowers due to quantum confinement effects and match the electron affinity of 2H MoS₂ as demonstrated by our density functional theory (DFT) results in Figure R1 below, which reduces the Schottky barrier height and thus the achievable contact resistance.

We added the DFT results and the related discussion as Fig. S11 in page 14 of the revised supplementary materials, for the reference of interested readers.

Figure R1: DFT-calculated work function with the varied width of 1T MoS₂ nanoribbon.

Encapsulating BN has been used previously to reduce extrinsic scattering (e.g. Nature Nanotech. 10, 534-540, 2015), but the technique reported in this work enables efficient contact. The O₂ plasma was also extensively exploited in TMD device fabrication (e.g. etching in 2D Mater. 3, 021007, 2016) and functionalization (Science advances 4, eaao5031, 2018), while this work establishes an innovative approach to realizing bond distortion in a controlled way to enhance the contact. Some other methods aim to reduce the damage introduced in metal deposition (e.g. Nature 557, 696-700, 2018; Nano Letters, 1416, 2018), while none of them deals with the orbital hybridization - perhaps the authors can comment on the metal deposition method they used.

Response: Our metal deposition was performed by Peva 450E using the non-destructive metal deposition parameters according to the reference (Nature 568, 70-74, 2019), i.e. critically high vacuum of 3×10^{-7} Torr and ultraslow deposition rate of 0.2 Å/s, right after the oxygen-plasma treatment to minimize the influence from the metal deposition process.

As pointed out by the reviewer, the mentioned two papers (Nature 557, 696-700, 2018; Nano Letters, 1416, 2018) reported the van der Waals metal-semiconductor integration to reduce the damage introduced by the contact metal deposition. Since their metal electrodes are deposited on the sacrificial substrate (polished Si) and mechanically transferred onto channel materials while mounted in either polymer or BN, respectively, their metal deposition does not require the non-destructive conditions (high vacuum or slow rate). And none of them deals with the orbital hybridization, which lies in the core idea of our contact strategy.

We improved the details of our metal deposition method in page 19 line 15-18 of the revised manuscript as “The non-destructive metal deposition²², i.e. critically high vacuum of 3×10^{-7} Torr and ultraslow deposition rate of 0.2 Å/s, was performed by Peva 450E right after the oxygen-plasma treatment to minimize the influence from the metal deposition process and seal the contact windows.”.

Considering these merits, this referee considers this work represents a substantial technological advance over those previous studies for enhancing TMD transistors.

However, the referee has some minor concerns:

- As the key techniques, the RIE etching and plasma treatment are not described with sufficient details. The methodologies given in the manuscript cannot be followed and reproduced by other labs. The effectiveness of these methods are not described sufficiently in the manuscript. To demonstrate the reproducibility and practicality of the method, it is crucial to establish the controllability and precision of these methods.

Response: Thanks for the significant suggestion. The RIE etching and plasma treatment were performed using the RIE system in a combination of parallel plate and inductively coupled plasma (ICP), model STS Pro, under the conditions as tabulated in detail below. The ICP can generate a high density source of ions while the separate RF bias in parallel plate can create directional electric fields near the substrate. In our soft oxygen plasma treatment, the vertical substrate bias is tuned to ~0 to reduce the kinetic energy of oxygen ions for complete reaction with samples and also to avoid possible etching effects.

Process	RF power & frequency	Gas composition	Pressure & flow rate	Substrate bias	Duration
Etching	200W 13.56MHz	CHF ₃ : O ₂ = 10 : 1	Base pressure 6 mTorr; Flow rate 40 : 4 sccm	20W	*5-15s
Plasma treatment	100W 13.56MHz	O ₂ 100%	Base pressure 6 mTorr; Flow rate 40 sccm	0W	10s

*The etching duration depends on the thickness of top BN layer in the top-contact geometry or BN/TMDSC/BN stack in the edge-contact geometry.

The plasma treatment is reproducible and practical once suitable conditions are set up. As shown in Figure R2, the contact resistance and field-effect mobility of multiple back-gated 3L-MoS₂ and 5L-WSe₂ FETs using the LBD contacts show similar high quality. The device deviation is reasonable considering the fabrication error by human handling and laboratory facilities. We notice that WSe₂ FETs show a larger device-to-device variation, which may be attributed to the higher mixed state of octahedral derivatives in LBD-treated WSe₂ than the situation of MoS₂.

Figure R2: Statistical analysis of contact resistance and carrier mobility of a number of LBD-contacted TMDSC FETs, showing the reproducibility and practicality of our method.

Therefore, we included the detailed RIE parameters in page 19 line 19-28 and the discussion of method reproducibility and practicality in page 20 line 1-7 of the revised manuscript. The Figure R2 is presented as Fig. S8A in page 11 of the revised supplementary materials.

- Raman results need further elaboration and clarification - spot size etc. Please explain how the 1nm distortion would cause such significant changes.

Response: Thanks for the good question. The Raman results were not collected from the edge contact, instead, as shown in Figure R3, we measured the Raman spectra during the fabrication of top contacts, which allow multiple phase-transitioned areas of around 1.8x2.3 μ m² of each for the spectroscopy. Although the phase-transitioned depth is only 1nm, the summed-up lateral area of multiple contact windows should be enough to generate the characteristic vibrational peaks of LBD. We conducted the Raman experiment using a micro Raman system (model

InVia of Renishaw), whose spot size is about 5 μm in diameter. The laser power during measurements was kept below 100 μW to avoid any possible sample damage and excessive heating. The silicon Raman mode at 520 cm^{-1} was used for the system calibration before spectra acquisition. Long-time ($\sim 30\text{s}$) signal acquisition at each $\sim 1.8 \times 2.3 \mu\text{m}^2$ phase-transitioned contact window was adopted and further integrated from six contact windows in the same device to produce the final spectra of higher signal-to-noise ratio.

To clarify our Raman measurement, we improved the detailed description in page 5 line 9-12 and page 6 line 1-4 of the revised supplementary materials accordingly.

Figure R3: Optical image of the $\sim 1.8 \times 2.3 \mu\text{m}^2$ plasma-treated contact windows in a top-contact-fabricated device for Raman measurements.

- Are there any observable distortions in the second layer of the WSe_2 sample? Please comment on this and explain its significance.

Response: Thank you for the question. We do not observe detectable distortion in the second layer of the WSe_2 sample as shown in Figure R4. The variation of distortion across a wider range and in different layers of WSe_2 is quantified by calculating the lattice parameter, a , from projected spacing of each unit cell. It can be seen that the bond distortion starts right inside the contact region, as marked by orange arrows, showing locally shrunk and elongated unit-cell sizes. But the second layer is protected by the topmost layer against the oxygen plasma interactions and its unit-cell size remains relatively constant as that in the standard structure.

The significance of such single-layer-depth distortion lies in the controllability and generality of the soft oxygen plasma treatment for monolayer and few-layer samples, since the LBD self-saturates in the topmost layer and serves as the carrier-injection bridge no matter of the sample thicknesses.

To clarify this issue, we improved the single-layer-depth discussion in page 9 line 8-10 of the revised supplementary materials as “Such single-layer-depth distortion explains the good controllability of the soft oxygen plasma treatment from monolayer to few-layer samples, since the LBD self-saturates in the topmost layer and serves as the carrier-injection bridge no matter of the sample thicknesses.”. The Figure R4 was presented as Fig. S6C in page 8 of the revised supplementary materials.

Figure R4: Bond distortion plot of the topmost and second layers in the WSe_2 contact site. The upper and middle panels are low- and high-magnification images of the contact region of top-contacted WSe_2 , respectively. The red and green dots in the middle panel indicate the fitted atomic positions of W and Se atoms.

- The role of oxygen in the treated samples need to be clarified.

Response: Thank you for allowing us to clarify this issue. On one hand, the oxygen plasma transfers kinetic energy to treated samples to drive the bond distortion; On the other hand, the generated oxygen ions can substitute some chalcogen atoms as demonstrated by our EELS analysis in Fig. S3 and thus locally modifies the bond length, stabilizing the distorted structure. It is also reported that the incorporation of oxygen species may increase the carrier transmission in the electrical interface with metals (Science 2013, 342, 6158, 614-617; Science Advances 2018, 4, eaao5031).

We emphasize the role of oxygen by modifying the related discussion in page 9 line 6-9 of the revised manuscript as “On one hand, the oxygen plasma transfers kinetic energy to treated samples to drive the bond distortion; On the other hand, the generated oxygen ions can substitute some chalcogen atoms during the soft O_2 -plasma treatment and thus locally modifies the bond length, stabilizing the distorted structure.”, and line 11-12 of the same page as “It is also reported that the incorporation of oxygen species may increase the carrier transmission in the electrical interface with metals.^{29,30}”.

Reviewer #2 (Remarks to the Author):

Report for “Bridging the gap between atomically thin semiconductors and metal leads” by X. Cai et al. The authors report on local bonding distortion (LBD) for good contacts between TMDSCs and metals. The impressive performances of the authors’ devices, such as the ultralow contact resistance (down to 90 ohm μm) and ultrahigh FET mobility (up to $358,000 \text{ cm}^2\text{V}^{-1}\text{s}^{-1}$), could be reported somewhere for 2D community, but this reviewer does not recommend this manuscript for publication in Nature Communications by the following reasons.

1. Novelty issue: as the authors wrote, microscale phase transformation (ref. 12, 13) has been reported for contact engineering. Plasma-treated phase engineering for contact has been also reported in other journals (not cited in this manuscript, for example JACS 139, 10216 (2017)). The authors claimed, “It is also different from the common H-to-T phase transition in TMDSCs by its nanoscale localization.” in page 5, but all the contact engineering would include randomly ordered nanoscale localization (or atomically defined contact between TMDSCs and metals) and edge structures. Thus, this work could be considered an extension of previous works, and this reviewer judges that the manuscript lacks the critical novelty for nature communications.

Response: Thank the opportunity for us to clarify the novelty of our work. Regarding the microscale phase transformation, the work function of bulk 1T MoS_2 mismatches the electron affinity of 2H MoS_2 , leading to a Schottky-barrier-limited contact and larger contact resistance (Phys. Rev. B 96, 165402, 2017). When the size of 1T phase decreases ($\sim 1 \text{ nm}$ in our cases), its work function lowers and match the electron affinity of 2H MoS_2 as demonstrated by our density functional theory (DFT) results in Figure R1 below, which reduces the Schottky barrier height and achievable contact resistance.

As for the plasma method, the mentioned JACS reported the 1H-to-1T phase transition in 1L MoS_2 using argon plasma, which is indeed a pioneering work of the contact improvement by plasma treatment. But the energetic argon bombardment actually creates many S vacancies to stabilize the phase transition, which is different from our structurally seamless soft oxygen plasma method. The vacancy-included 1H-1T interface is not applicable to achieve ultrahigh mobility and explore quantum transport phenomena, where the in-gap energy levels from vacancy and dangling bonds will result in Fermi level pinning and Schottky effects, especially at cryogenic temperature (Physical Review Letters 84, 6078, 2000; ACS Nano 11, 1588, 2017).

Our local bond distortion (LBD) strategy is distinct from these literature reports. The bond distortion is not the conventional 1T phase transition or disorder, but an intermediate structure towards T phase without defects (no vacancies or dangling bonds). Such LBD is stabilized through the oxygen substitution of partial chalcogen atoms. Our term of “nanoscale localization” may not be so accurate, where we were trying to explain the nanoscale size of LBD and resulted matched work function. To clarify the novelty of this work, our work establishes an innovative approach to realizing bond distortion in a highly controlled way, enhancing the orbital hybridization and resulted carrier-injection efficiency in metal-TMDSC contacts, which have not been dealt with in previous reports.

Therefore, we added the DFT results and the related discussion as Fig. S11 in page 14 of the revised supplementary materials, for the reference of interested readers. We included the JACS paper as ref. 14 in page 2 line 14 of the revised manuscript and emphasized the novelty of this work from the literature works in page 3 line 15-22 as “We note that there is pioneering contact improvement works by argon-plasma-induced T phase,¹⁴ but the energetic argon bombardment

actually creates vacancies to stabilize the phase transition, which is different from our structurally seamless soft oxygen plasma method. And our LBD is not the conventional 1T phase transition, but an intermediate structure towards tetragonal phases without vacancies or dangling bonds. Our work establishes an innovative approach to enhancing the orbital hybridization and carrier-injection efficiency in metal-TMDSC contacts by a highly controlled way, which have not been dealt with in previous reports.”. We also changed the term “nanoscale localization” in page 5 line 15 to “nanoscale sizes and tuned work functions (Fig. S11)” to improve the language accuracy.

Figure R1: DFT-calculated work function with the varied width of 1T MoS₂ nanoribbon.

2. Concerning the nanoscale localization mentioned in page 5, this reviewer believes that Raman spectroscopy shown in Fig. S4 should not be observed in the 1 nm-wide edge in Fig. 3. The authors better add more explanations.

Response: Thanks for the chance to explain our Raman experiment. The Raman results shown in Fig. S4 were not from the edge-contacts in Fig. 3 of the main text. Instead, as shown in Figure R3, we measured the Raman spectra during the fabrication of top contacts, which allow multiple phase-transitioned areas of around $1.8 \times 2.3 \mu\text{m}^2$ of each for the spectroscopy. Although the phase-transitioned depth is only 1nm, the summed-up lateral area of multiple contact windows is enough to generate the characteristic vibrational peaks of LBD. We conducted the Raman experiment using a micro Raman system (model InVia of Renishaw), whose spot size is about $5 \mu\text{m}$ in diameter. The laser power during measurements was kept below $100 \mu\text{W}$ to avoid any possible sample damage and excessive heating. The silicon Raman mode at 520 cm^{-1} was used for the system calibration before the spectra acquisition. Long-time ($\sim 30\text{s}$) signal acquisition at each $\sim 1.8 \times 2.3 \mu\text{m}^2$ phase-transitioned contact window was adopted and further summed up from six contact windows in the same device to produce the final spectra of higher signal-to-noise ratio.

To clarify our Raman measurement, we improved the detailed description in page 5 line 9-12 and page 6 line 1-4 of the revised supplementary materials accordingly.

Figure R3: Optical image of the $\sim 1.8 \times 2.3 \mu\text{m}^2$ plasma-treated contact windows in a top-contact-fabricated device.

3. The contact resistance of ‘90 ohm μm ’ needs more supporting data. At zero gate bias, the current should be at the ‘off-state’ (e.g. we see such ‘off state’ in Fig. S7C). However, the y-axis intersection is 500 ohm, which is only five times the value at $V_g=60 \text{ V}$. The lack of ‘off-state’ at zero gate bias may indicate some technical issues in the measurement.

Response: We show the full range of extracted contact resistance from this edge-contacted 3L-MoS₂ in Figure R5 below. The off-state of this device locates below -10 V back gate voltage as demonstrated in Figure R5, as well as in the inset of Fig. 4D, where the complete metal-insulator transition behaviors at varied temperatures are depicted. This is not a technical issue in the measurement, but indicating the heavier intrinsic electron doping in MoS₂ than that in WSe₂ (see WSe₂ data in Fig. S7C). The shift of off-state to negative gating in n-type MoS₂ FETs is common because MoS₂ crystals are naturally electron-doped, such as Nature Materials 2014, 13, 1128 (off-state below -30V back gate voltage); JACS 2017, 139, 10216 (off-state below -30V back gate voltage); Nature Communications 2019, 10, 611 (off-state below -40V back gate voltage); Nature Electronics 2019, 2, 17 (off-state below -60V back gate voltage); Nature 2019, 568, 70 (off-state below -10V back gate voltage), to name a few similar examples. The transistors using different MoS₂ crystals may show different ‘off-state’ positions due to varied intrinsic doping levels originated from their synthetic conditions.

As shown in Figure R6, the contact resistance plot is extracted from the resistance difference between two-probe and four-probe measurements according to $2 * R_c = (R_{2p} - \frac{L_{out}}{L_{in}} R_{4p}) * W$, where R_{2p} and R_{4p} are the two-probe and four-probe resistance, respectively; L_{out} and L_{in} are the distance of outer probe pair and inner probe pair in the four-probe configuration, respectively. It can be seen that the contact resistance depends little on the back gating, i.e. the carrier density, after the device is turned on fully ($V_{bg} > 20\text{V}$) and the contact resistance of 90 $\Omega\mu\text{m}$ in the edge-contacted 3L-MoS₂ FET is achieved under the conditions of 60V back-gate voltage, i.e. $\sim 6 \times 10^{12} \text{ cm}^{-2}$ carrier density.

To clarify the 90 $\Omega\mu\text{m}$ contact resistance, we replaced the Fig. 4C of the revised manuscript by Figure R6 and modified the caption accordingly in page 10 line 3-7 as “(C) Contact resistance, R_c , extracted from the resistance difference between two-probe and four-probe measurements as the function of gate voltages. It can be seen that the contact resistance depends little on the back gating after the device is fully turned on. The off-state of this device locates below -10 V back gate voltage due to the natural electron doping of MoS₂ as demonstrated in the inset of Fig. 4D.”.

Figure R5: The extracted contact resistance of edge-contacted 3L-MoS₂ in the full back gating range, showing the off-state region (blue shaded).

Figure R6: The contact resistance extraction of edge-contacted 3L-MoS₂ FETs.

4. The authors claimed that the slight deviation of atomic position makes an ideal contact like Fig. 1D. This is contrasted with our intuition. Such disordered atomic arrangements normally make Fermi level pinning to fix a Schottky barrier. Accepting the authors' explanation, how could the atomic deviation be well-adjusted for such an ideal contact? This might be an extraordinary coincidence. One possible explanation would be that a (vdW) tunneling barrier is formed by oxygen compounds for the improved contact.

Response: Thanks for the question. In Fig. 1D of the main text, we use the schematic to show that the semi-metallic distorted phase can generate dispersed energy states around the electron affinity of channel semiconductor, benefiting the electron transmission from metal leads to channels. And the work function of the semi-metallic LBD can be tuned by its width to match the band edge of channel semiconductor as shown by Figure R1 above. So the band diagram is an ideal possible situation.

Fermi level pinning and Schottky barrier are closely connected phenomena, originating from dangling bonds of semiconductors or surface dipoles induced by metal leads (Physical Review Letters 2000, 84, 6078; ACS Nano 2017, 11, 1588). However, our LBD is not a disordered structure with defects (vacancies or dangling bonds), but an intermediate distorted tetragonal phase with semi-metallic properties. Such LBD is stabilized through the oxygen substitution of partial chalcogen atoms and improves the carrier transmission efficiency at the electrical interface (Science 2013, 342, 6158, 614-617; Science Advances 2018, 4, eaao5031).

As for the suspicion of oxygen compounds as a vdW tunneling barrier, we do not observe such structure in our cross-section observation of practical contact interfaces. And importantly the oxygen-plasma treated MoS₂ is metallic as shown in the Fig. S10 of supplementary materials and connected directly to metal leads in edge contacts, thus the leads-LBD interface cannot provide the tunneling barrier for van der Waals tunneling mechanisms. Therefore, although we appreciate the idea suggested by the reviewer and believe it may be a new route to improve the metal-TMDSC contacts, it is unlikely to be the case here.

5. The graphene edge contact has been found to be about ‘100 ohm um’, and the edge contact in this work shows a better value than the graphene result. Given that the edge contact coupling length in graphene is shorter (~1 Å) than that of MoS₂ in this work (1.8 Å), the upper bound (90 ohm um) in this work is too low. The authors show a Hall bar geometry in Fig. 5A. Have the authors conducted 4-probe measurements to precisely estimate the contact resistance?

Response: Thanks for the good question. We agree with the reviewer’s concern about the upper bound of contact resistance from the perspective of “edge contact coupling lengths”, but the comparison is in fact not appropriate. Because the achievable contact resistance of 2D materials is strongly layer-number-dependent as shown in Fig. S8B of our supplementary materials. For the same device structure and LBD contact method, thinner channel leads to a larger contact resistance. This trend is also effective for graphene, i.e. 100 Ωμm in single-layer graphene FETs, 11 Ωμm in 2L graphene FETs and ~9 Ωμm in 3L graphene FETs (ACS Nano 2014, 8, 1, 994). Therefore, our 90 Ωμm contact resistance of 3L-MoS₂ should be compared with the case of ~9 Ωμm in 3L graphene. Our 3L-MoS₂ contact resistance is still one-order larger than that in 3L-graphene. The estimation of 3L MoS₂ contact resistance upper bound by “the edge contact coupling length” should refer to the value of ~9 Ωμm in 3L graphene.

Yes, the contact resistance of 90 Ωμm is indeed estimated by 4-probe measurements and the original 2-probe description of Fig 4C is made by mistake. Here we sincerely apologize for this carelessness and have corrected it in the revised version in page 10 line 3-7 as “(C) Contact resistance, R_c , extracted from the resistance difference between two-probe and four-probe measurements as the function of gate voltages. It can be seen that the contact resistance depends little on the back gating after the device is fully turned on. The off-state of this device locates below -10 V back gate voltage due to the natural electron doping of MoS₂ as demonstrated in the inset of Fig. 4D.”. We provide the complete data in Figure R6. The contact resistance plot is extracted from the resistance difference between two-probe and four-probe resistivity according to $2 * R_c = (R_{2p} - \frac{L_{out}}{L_{in}} R_{4p}) * W$, where R_{2p} and R_{4p} are the two-probe and four-probe resistance, respectively; L_{out} and L_{in} are the distance of outer probe pair and inner probe pair in the four-probe configuration, respectively. The contact resistance of 90 Ωμm in the edge-contacted 3L-MoS₂ FET is achieved under the conditions of 60V back-gate voltage, i.e. ~6 x 10¹² cm⁻² carrier density.

As for the contact resistance extraction by 4-probe measurements of the device in Fig. 5A, which is a top-contacted WSe₂ FET, is shown in Fig. S7A of supplementary materials.

Figure R6: The contact resistance extraction of edge-contacted 3L-MoS₂ FETs.

Reviewer #3 (Remarks to the Author):

Cai et al. reported on the fabrication and characterization of Hall bar-like TMDs (2H-MoS₂ and 2H-WS₂) field-effect devices with low apparent contact resistance. The authors claimed that the oxygen plasma treatment of the contact areas immediately before metallization results in the contact resistance lower than the best values reported so far (see Nature 593, 211 (2021)). The reduction of the contact resistance is attributed to the formation of the highly electrically conducting 1T phase of TMD, presumably under mild plasma treatment. The presence of lattice distortions in the contact area, which resembles the 1T phase, was observed by the TEM data for both materials used in the study.

Similarly to the previous reports (Nature Mater. 13, 1128 (2014)), the authors attributed the low apparent contact resistance, found from the two-terminal measurements at the highest possible charge carrier concentration, to the 1T-like lattice distortions expected to have metal-like electrical conductivity. Although the idea of crystal phase engineering to achieve low contact resistance is not new, the fabrication approach involving plasma treatment reported by Cai et al. is more practical and has significant technological potential.

However, in my opinion, the claims made in this work are very premature, and much additional work has to be done to prove the relation between the plasma-induced lattice distortions and contact resistance as well as its actual value. Below are my comments and recommendations concerning this work:

1) The proper contact resistance measurement is absent from this work. The transmission line experiment should be carried out for each contact geometry (edge-contact and top-contact) as a function of channel carrier concentration and temperature. It is insufficient to quote the two-terminal channel resistance at 2 Kelvin to estimate the contact resistance in the entire range of possible parameters. A separate assessment of the situation is also needed for the p-type channel material, such as MoSe₂ or WSe₂.

Response: Thank you for the useful suggestion. We conducted transmission line experiments on n-type MoS₂ using LBD edge contacts, n-type and p-type WSe₂ using LBD top contacts as

shown in Figure R7 below. The contact resistance values from extrapolated intercepts at varied channel carrier concentrations and temperatures for different device types are tabulated below. It can be seen that the obtained best contact resistance is $85 \pm 15 \text{ } \Omega \mu\text{m}$ in edge-contacted 3L-MoS₂, which is very close to our estimation of $90 \text{ } \Omega \mu\text{m}$. The extrapolated negative values of contact resistance in the edge-contacted MoS₂ devices at room temperature are attributed to the large fitting uncertainty of such tiny values and the thermal-excited orbital hybridization states in edge contacts. For n-type top-contacted WSe₂, the contact resistance falls within 100-200 $\Omega \mu\text{m}$ across varied carrier densities and temperatures. In general, the contact resistance is dependent on the carrier density and the temperature. The higher carrier density and lower temperature lead to the better contact resistance, which may be explained by one of facts that the impurity scattering in channels or contact interfaces can be screened by increased carrier density and suppressed by lowered temperature. This correlation in top-contacted p-type WSe₂ is more prominent, which may be due to the weak interaction between LBD and Pd leads.

Device Type	$n=1 \times 10^{12} \text{ cm}^{-2}$ 2K	$n=4.2 \times 10^{12} \text{ cm}^{-2}$ 2K	$n=1 \times 10^{12} \text{ cm}^{-2}$ 300K	$n=4.2 \times 10^{12} \text{ cm}^{-2}$ 300K
n-type MoS ₂ (edge-contact)	$328 \pm 61 \text{ } \Omega \mu\text{m}$	$85 \pm 15 \text{ } \Omega \mu\text{m}$	$-88 \pm 85 \text{ } \Omega \mu\text{m}$	$-32 \pm 99 \text{ } \Omega \mu\text{m}$
n-type WSe ₂ (top-contact)	$147 \pm 56 \text{ } \Omega \mu\text{m}$	$155 \pm 24 \text{ } \Omega \mu\text{m}$	$271 \pm 97 \text{ } \Omega \mu\text{m}$	$167 \pm 155 \text{ } \Omega \mu\text{m}$
p-type WSe ₂ (top-contact)	$784 \pm 256 \text{ } \Omega \mu\text{m}$	$158 \pm 47 \text{ } \Omega \mu\text{m}$	$1679 \pm 470 \text{ } \Omega \mu\text{m}$	$1076 \pm 437 \text{ } \Omega \mu\text{m}$

Therefore, we included the transmission line experiment results as Fig. S12 in page 15 of the revised supplementary materials, and discussed the results in page 14 line 26-31 and page 15 line 1-4 of the revised manuscript as “We also conducted transmission line experiments on n-type MoS₂ using LBD edge contacts, n-type and p-type WSe₂ using LBD top contacts as shown in Fig. S12. It can be seen that the obtained best contact resistance is $85 \pm 15 \text{ } \Omega \mu\text{m}$ in edge-contacted 3L-MoS₂, which is very close to our estimation of $90 \text{ } \Omega \mu\text{m}$. For n-type top-contacted WSe₂, the contact resistance falls within 100-200 $\Omega \mu\text{m}$ across ranges of carrier densities and temperatures. In principle, the contact resistance is dependent on the carrier density and the temperature. The higher carrier density and lower temperature lead to the better contact resistance, which may be explained by the fact that the impurity scattering in channels or contact interfaces can be screened by clouded carriers and suppressed by lowered temperature. This correlation in top-contacted p-type WSe₂ is more prominent.”. We also added the paper Nature 593, 211 (2021) as ref 31 in page 14 line 26 because it gives us useful guidance on the proper contact resistance extraction.

Figure R7: (A) Optical image of a transmission line device. Transmission line measurements of (B) edge-contacted n-type MoS₂, (C) top-contacted n-type WSe₂ and (D) top-contacted p-type WSe₂ at varied channel carrier densities and temperatures.

2) Regrettably, the authors did not disclose the details of the 1T phase preparation procedure (soft oxide plasma treatment), which seems to be the main novelty of this work. For this work to be publishable, the authors must ensure that the description of the plasma treatment procedure is discussed in the smallest detail to be reproduced by the others. In particular, the following information should be provided:

- model and type of the RIE plasma system,
- RF power and frequency,
- gas mixture composition,
- pressure and flow rate,
- observed substrate bias,
- treatment duration.

Any other details/tips/hints should also be presented.

Response: Thanks for the significant advice. The soft oxygen plasma treatment was performed using the RIE system in a combination of parallel plate and inductively coupled plasma (ICP), model STS Pro, under the conditions as tabulated in detail below. The ICP can generate a high density source of ions while the separate RF bias in parallel plate can create directional electric

fields near the substrate. In our soft oxygen plasma treatment, the vertical substrate bias is tuned to ~0 to reduce the kinetic energy of oxygen ions for their complete reaction with samples and also to avoid possible etching effects.

Process	RF power & frequency	Gas composition	Pressure & flow rate	Substrate bias	Duration
Plasma treatment	100W 13.56MHz	O ₂ 100%	Base pressure 6 mTorr; Flow rate 40 sccm	0W	10s

Therefore, we included the detailed RIE parameters in page 19 line 19-28 of the revised manuscript.

3) Some additional work with TEM should be done to establish the mechanism of plasma-induced structural transition and eliminate the possible experimental artefacts caused by the e-beam irradiation. For example, Lin et al. (Nature Nanotech. 9, 931 (2014)) reported the e-beam induced 2H-to-1T transition in MoS₂. So, one has to ensure that the observed lattice distortion results from the plasma treatment and is not triggered by the e-beam exposure during TEM imaging.

Response: Thanks for the good question. In the mentioned paper (Nature Nanotechnology 2014, 9, 391), both heating to a high temperature (600°C) and applying ultrahigh electron dose (>40 MeV nm⁻²) are necessary for the 1H-to-1T phase transition in single-layered MoS₂. But we conducted the TEM imaging at room temperature and using a low electron dose of ~7.5-20.0 Me nm⁻², of which conditions are far away from the threshold to cause electron-induced structural transitions.

We further exclude the possible experimental artefacts caused by the e-beam irradiation by the control experiments on pristine MoS₂ monolayers. The imaging conditions are the same as used for the previous imaging of device contact regions. As shown in Figure R8, the 2H structure can be unambiguously observed without any structural distortion or experimental artefacts under our e-beam irradiation conditions, both along the plane-view [001] and the cross-section [100] directions. Therefore, the observed lattice distortion is not triggered by the e-beam, and should be originated from the plasma treatment.

Therefore, we included the additional TEM work as Fig. S13 to exclude e-beam artefacts in page 16 of the revised supplementary materials, and discussed the results in page 8 line 17-21 of the revised manuscript as “Since the atomic structure on the left H region remains identical with the simulated one, there should be no damage or artefact introduced by our observation. The safeness of our imaging conditions to exclude any e-beam artefacts is also demonstrated by the controlled imaging of pristine monolayer MoS₂ under the same conditions as the atomic imaging of device contact regions (Fig. S13).”. We also discussed the mentioned paper (Nature Nanotechnology 2014, 9, 391) in page 21 line 11-14 of the revised manuscript as “Our imaging conditions of low acceleration voltage of 60 kV, low electron irradiation dose of ~7.5-20.0 Me nm⁻² and room temperature remain far away from the threshold to cause electron-induced structure change.³²”.

Figure R8: (A) Plane-view and (B) cross-section imaging of the pristine MoS₂.

4) Apart from the essential contact resistance characterization, a more detailed transport study of the Hall bar devices needs to be done. On the one hand, it seems that the WS₂ device with nominally higher carrier mobility possesses minimal signs of quantum oscillations. On the other, a more modest MoS₂ device exhibits well-pronounced QHE steps. This inconsistency has to be explained.

Response: Thank you for the comment. We show more detailed transport data of the Hall bar WSe₂ devices in Figure R9 below. Our high-carrier-mobility WSe₂ devices exhibit prominent quantum oscillations as demonstrated by the R_{xx} map in Figure R9A. The lowest integer filling factor 3 of such Q-valley system can be reached under a common 13T magnetic field by dilute electron gas shown in Figure R9B, proving the high quality of LBD-contacted devices. But the detailed analysis of these quantum transport physics and symmetry breaking behaviors is a bit beyond the contact topic of this paper. We plan to report a detailed transport study, including the high-magnetic-field (50T) results in another manuscript in the near future.

As for the concern about “inconsistency between carrier mobility and quantum oscillations”, the observed quantum phenomena and carrier mobilities are actually consistent. The quantum hall effect (QHE) not only correlates with the carrier mobility, but also requires the strict two-dimensional confinement of charge carriers. Unfortunately, the conduction band edge of 5L

WSe₂ locates at the Q valley, whose density of states are not confined within single atomic layers, but hybridized across the whole thickness (Reviews of Modern Physics 1986, 58, 519; Chemical Society Reviews 2015, 44, 2643). Thus, the 5L WSe₂ device manifests less pronounced QHE steps than the 1L MoS₂ (highly 2D confined K valley). In contrast to the QHE, the Shubnikov-de Haas (SdH) oscillation turns out to be a quantum phenomenon without the requirement of two-dimensional edge states and its onset in the magnetic field (B_q) is a good indicator of the lower-bound of carrier mobility prone for quantum phenomena, namely the quantum mobility ($\sim 1/B_q$). As demonstrated in Fig. 5E in page 12 of the manuscript, the 5L-WSe₂ device shows a quantum oscillation onset at a much lower magnetic field strength (~ 0.4 T, representing an ultrahigh quantum mobility of $\sim 25,000$ cm²V⁻¹s⁻¹) than that of 1L-MoS₂ device shown in Fig. S5 of the supplementary materials (~ 4 T, representing a quantum mobility of $\sim 2,500$ cm²V⁻¹s⁻¹), which is well consistent with their difference of carrier mobilities measured by the field and Hall effects, i.e. field-effect and Hall mobilities.

Therefore, we included the extra transport data and discussion in Figure R9 as Fig. S14 in page 17 of the revised supplementary materials for the interest of professionals. We also discussed the significance of quantum mobility as one of the device quality criteria in page 14 line 1-3 of the revised manuscript as “The quantum mobility is an evaluation of the system’s quality for observing quantum phenomena (see the more detailed quantum transport data of such high-mobility WSe₂ device in Fig. S14).”.

Figure R9: (A) Color map of the R_{xx} of 5L-WSe₂ FET, showing the Landau quantization along varied gate voltages (carrier density) and magnetic field strengths. (B) SdH oscillations at low carrier densities, showing the achievable lowest integer Landau level of Q valley.

Before this essential work is done, reporting the apparent low-contact resistance cannot be considered reliable. However, if the initial clues presented in the manuscript are proven correct, the impact of this work on the 2D material research and possible future application of TMDs in microelectronics will be significant.

Considering the above comments, I cannot recommend this work in its present form for publishing in either Nature Communication journal or any other Nature partner journal. I would strongly encourage the authors to continue working on this problem to prove the validity of the fabrication method by confirming the reproducible low-contact resistance to different materials and contact geometries and achieve a better understanding of its origin.

Response: Thank you for giving us the chance to improve this work. We hope the additional experiment results and the revised version now meet your expectations and the publication standard.

Other changes made to the revised manuscript:

1. The author list in page 1 line 4 of the revised manuscript was modified by adding Yong Chen as a coauthor, in recognition of his contribution to the extra data provided in this revision. All other authors agreed on the addition and the order adjustment of authorship.
2. The original reference labelling was accordingly adjusted in compliance with the insertion of new references.
3. The funding acknowledgements in page 23 line 2-4 of the revised manuscript was updated as “The work is partially supported by the National Key R&D Program of China (2020YFA 0309600), the Hong Kong Research Grants Council (Projects No. AoE/P-701/20, C7036-17W, 16303720 and C6025-19G) and the William Mong Institute of Nano Science and Technology.”.

REVIEWER COMMENTS

Reviewer #1 (Remarks to the Author):

Thanks for the responses. The authors have addressed my concerns properly and the manuscript is good enough for publication. I recommend acceptance.

Reviewer #2 (Remarks to the Author):

Report for "Bridging the gap between atomically thin semiconductors and metal leads" by X. Cai et al.

In the response letter, the authors addressed well this review's concerns in a clear and scientific way. The authors' answers make sense in the response letter.

This reviewer still believes (from his/her own experiences) that effective (O₂, 100 W) plasma treatments cannot produce such subtle, homogeneous, and disorder-free surfaces of 2D materials. All plasma processes are violent and destructive. Moreover, this reviewer thinks the oxygen-substitution-driven local structures would play a similar role (with the local dangling bonds or surface dipole) for Fermi level pinning in the transport.

Nevertheless, the authors show clear TEM images of some local areas to rebut this reviewer's concerns in the revision, which could be convincing for other readers. Thus, this reviewer has no objection for publication of this manuscript in Nature Communications.

Reviewer #3 (Remarks to the Author):

I am generally happy with the authors' responses except for the transmission line data. Obviously, the room temperature part of these data is inadequate due to the limited accuracy of the experiment. However, other reviewers requested a similar measurement performed by comparing the two- and four-terminal resistance. In this regard, it would be sufficient to add to Figure 5c an extra set of data taken at room temperature. Also, it seems like the data units presented in that figure are inconsistent: - R_c is measured in [$\mu\text{m}\cdot\text{Ohm}$] and R_{2p}, R_{4p} are in [Ohm]. The work can be published after the above comments are addressed.

(Note: reviewers' comments are copied verbatim here in black with our responses follow in blue; corresponding changes made to the manuscript or the supplementary materials are highlighted by cyan background in respective documents, which are also copied here in red for reference.)

Reviewer #1 (Remarks to the Author):

Thanks for the responses. The authors have addressed my concerns properly and the manuscript is good enough for publication. I recommend acceptance.

Response: Thank you so much!

Reviewer #2 (Remarks to the Author):

Report for “Bridging the gap between atomically thin semiconductors and metal leads” by X. Cai et al. In the response letter, the authors addressed well this review's concerns in a clear and scientific way. The authors' answers make sense in the response letter.

This reviewer still believes (from his/her own experiences) that effective (O₂, 100 W) plasma treatments cannot produce such subtle, homogeneous, and disorder-free surfaces of 2D materials. All plasma processes are violent and destructive. Moreover, this reviewer thinks the oxygen-substitution-driven local structures would play a similar role (with the local dangling bonds or surface dipole) for Fermi level pinning in the transport.

Nevertheless, the authors show clear TEM images of some local areas to rebut this reviewer's concerns in the revision, which could be convincing for other readers. Thus, this reviewer has no objection for publication of this manuscript in Nature Communications.

Response: Thank you for recommending the publication of this work!

Our device measurements and structural characterization of electrical interfaces prove consistently that the well-configured soft oxygen plasma treatment is not destructive, but triggers local bonding distortion (LBD) in 2D semiconductors to improve the carrier injection. No Fermi level pinning effects were observed in our LBD-contacted devices. Our work establishes an innovative approach to enhancing the orbital hybridization and carrier-injection efficiency in metal-TMDSC contacts by highly controlled oxygen plasma. We thus believe that this paper can advance the community's understanding on the oxygen plasma process and its advantages for high-performance electrical contacts of atomically thin semiconductors.

Reviewer #3 (Remarks to the Author):

Dear Editor,

I am generally happy with the authors' responses except for the transmission line data. Obviously, the room temperature part of these data is inadequate due to the limited accuracy of the experiment. However, other reviewers requested a similar measurement performed by comparing the two- and four-terminal resistance. In this regard, it would be sufficient to add to Figure 5c an extra set of data taken at room temperature.

Response: Thanks for the suggestion. Because our local bonding distortion (LBD) strategy makes ultrasmall contact resistance, the room-temperature (RT) transmission line data of MoS₂ shown in our previous revision seem less accurate for extracting such tiny contact resistance values. As the complementary dataset, in Figure R1, we demonstrate the room-temperature extraction of contact resistance of LBD-contacted MoS₂ and WSe₂ devices by comparing the two- and four-terminal resistance. The extracted R_c at room temperature falls below 200 $\Omega \mu\text{m}$ when the device is fully turned on ($V_{bg} > 40\text{V}$). The slightly larger room-temperature R_c than the low-temperature case (e.g. $\sim 90 \Omega \mu\text{m}$ of MoS₂) is attributed to the strong phonon scattering in the metal-TMDSC interfaces at room temperature.

Since the mentioned Figure 5c is actually a plot of tow-probe I_{ds} - V_{ds} curves at varied temperatures (from 0.3K to RT), we propose to add this complementary RT measurement, i.e. Figure R1, after the transmission line data as Fig. S15 in page 18 of the revised supplementary materials and improved the discussion in page 14 line 30-31 and page 15 line 1-2 of the revised manuscript into “As a complementary measurement, in Figure S15, we demonstrate the contact resistance extraction of LBD-contacted MoS₂ and WSe₂ devices by comparing the two- and four-terminal resistance, showing consistent contact resistance values at room temperature.”

Figure R1: The room-temperature contact resistance extraction by comparing the two- and four-terminal resistance of LBD-contacted (A) MoS₂ and (B) WSe₂ FETs. The four-probe resistance is scaled by L_{out}/L_{in} as described in the Methods section.

Also, it seems like the data units presented in that figure are inconsistent: R_c is measured in $[\mu\text{m}\cdot\text{Ohm}]$ and R_{2p} , R_{4p} are in $[\text{Ohm}]$.

Response: Thanks for pointing out this issue. We corrected this unit inconsistency by converting “ Ω ” to “ $\Omega \mu\text{m}$ ” using corresponding geometry factors in the revised Fig. S12 in page 15 of the revised supplementary materials, which was also copied here as Figure R2 below.

Figure R2: Transmission line measurement results with consistent units.

The work can be published after the above comments are addressed.

Sincerely,

Response: Thanks, the advices improve this work a lot.

REVIEWERS' COMMENTS

Reviewer #3 (Remarks to the Author):

These are two final comments after which the paper can be accepted for publication.

1) For clarity, the data in Figure 4(c) (main text) and Figure S15 (S.I.) should be presented on the logarithmic scale along the Y-axis.

2) In these figures, I strongly recommend presenting the actual resistance data for R_c , R_{4p} , R_{2p} measured in [Ohm] along with additional Y-axes on the right-hand side of each panel, which will mark the corresponding contact resistance R_c measured in [Ohm.μm]. In its current form, the presented data is confusing.

(Note: reviewers' comments are copied verbatim here in black with our responses follow in blue; corresponding changes made to the manuscript or the supplementary materials are highlighted by magenta background in respective documents, which are also copied here in red for reference.)

Reviewer #3 (Remarks to the Author):

Dear Editor,

These are two final comments after which the paper can be accepted for publication.

1) For clarity, the data in Figure 4(c) (main text) and Figure S15 (S.I.) should be presented on the logarithmic scale along the Y-axis.

2) In these figures, I strongly recommend presenting the actual resistance data for R_c , R_{4p} , R_{2p} measured in [Ohm] along with additional Y-axes on the right-hand side of each panel, which will mark the corresponding contact resistance R_c measured in [$\text{Ohm}\cdot\mu\text{m}$]. In its current form, the presented data is confusing.

Sincerely

Response: Thanks for the two comments for Figure 4(c) and Figure S15. As shown below, 1) the Y-axes were modified into the logarithmic scale; 2) the actual resistance values in [Ohm] were indicated by adding additional Y-axes on the right-hand side, which were correlated with the original [$\text{Ohm}\cdot\mu\text{m}$] axes on the left-hand side through the geometry factors of each device channel. Figure 4(c) in the main text page 22 and Figure S15 in the supplementary materials page 18 were accordingly modified. The captions were also improved by adding some explanation of the scales "...and both resistance values in $\Omega\cdot\mu\text{m}$ and Ω units are plotted in the logarithmic scale." in page 22 line 5-6 of the revised manuscript and "..., where both resistance values in $\text{k}\Omega\cdot\mu\text{m}$ and $\text{k}\Omega$ units are plotted in the logarithmic scale." in page 18 line 4-5 of the revised supplementary materials, respectively.

Revised Figure 4(c): Y-axes in the logarithmic scale, together with the extra Y-axis indicating the resistance values in the Ohm (Ω) unit.

a

b

Revised Figure S15: Y-axes in the logarithmic scale, together with the extra Y-axes indicating the resistance values in the Ohm ($k\Omega$) unit.